# Mental Health, Information and Being Connected: Qualitative Experiences of Social Media Use during the COVID-19 Pandemic from a Trans-National Sample

**DOI:** 10.3390/healthcare9060735

**Published:** 2021-06-15

**Authors:** Mariyana Schoultz, Janni Leung, Tore Bonsaksen, Mary Ruffolo, Hilde Thygesen, Daicia Price, Amy Østertun Geirdal

**Affiliations:** 1School of Health and Life Sciences, Northumbria University, Newcastle upon Tyne NE1 8ST, UK; 2School of Psychology, The University of Queensland, St. Lucia, QLD 4072, Australia; j.leung1@uq.edu.au; 3Department of Health and Nursing Science, Faculty of Social and Health Sciences, Inland Norway University of Applied Sciences, 2406 Elverum, Norway; tore.bonsaksen@inn.no; 4Faculty of Health Studies, VID Specialized University, 4306 Sandnes, Norway; 5School of Social Work, University of Michigan, Ann Arbor, MI 48109, USA; mruffolo@umich.edu (M.R.); daiciars@umich.edu (D.P.); 6Faculty of Health Sciences, Oslo Metropolitan University, 0167 Oslo, Norway; hilthy@oslomet.no; 7Faculty of Social Sciences, Oslo Metropolitan University, 0167 Oslo, Norway; amyoge@oslomet.no

**Keywords:** social media, COVID-19, cross-sectional, trans-national, mental health, loneliness, pandemic

## Abstract

*Background*: Due to the COVID-19 pandemic and the strict national policies regarding social distancing behavior in Europe, America and Australia, people became reliant on social media as a means for gathering information and as a tool for staying connected to family, friends and work. This is the first trans-national study exploring the qualitative experiences and challenges of using social media while in lockdown or shelter-in-place during the current pandemic. *Methods*: This study was part of a wider cross-sectional online survey conducted in Norway, the UK, USA and Australia during April/May 2020. The manuscript reports on the qualitative free-text component of the study asking about the challenges of social media users during the COVID-19 pandemic in the UK, USA and Australia. A total of 1991 responses were included in the analysis. Thematic analysis was conducted independently by two researchers. *Results*: Three overarching themes identified were: Emotional/Mental Health, Information and Being Connected. Participants experienced that using social media during the pandemic amplified anxiety, depression, fear, panic, anger, frustration and loneliness. They felt that there was information overload and social media was full of misleading or polarized opinions which were difficult to switch off. Nonetheless, participants also thought that there was an urge for connection and learning, which was positive and stressful at the same time. *Conclusion*: Using social media while in a shelter-in-place or lockdown could have a negative impact on the emotional and mental health of some of the population. To support policy and practice in strengthening mental health care in the community, social media could be used to deliver practical advice on coping and stress management. Communication with the public should be strengthened by unambiguous and clear messages and clear communication pathways. We should be looking at alternative ways of staying connected.

## 1. Introduction 

Since the Spanish flu over a century ago, the COVID-19 pandemic is the greatest and possibly the deadliest threat to public health worldwide [1,2]. The pandemic triggered new national policies on public behavior in most countries throughout Europe, America and Australia. Social distancing became the main public behavior policy apart from hygiene rules [1,3].

The severity and scale of the social distancing policy were exceptionally strict and included severe restrictions in many aspects of daily living for the majority of the population. Some of the restrictions were that people should stay at home in order to minimize contact and spread of the disease and leave their homes only in exceptional circumstances. In addition, nurseries, schools and universities were closed, with online study becoming the new norm overnight. Flights and non-essential travels were canceled, as well as sports, religious and cultural events. Non-essential businesses that require physical contact were also closed. Thus, the majority of people in Europe, the United States of America (USA) and Australia were urged into a lockdown, shelter-in-place or a quarantine (see Table A1 in Appendix A).

While there is evidence that a quarantine can be a useful measure to prevent the spread of communicable diseases [4], the psychological impact of a lockdown and shelter-in-place is immense. The negative psychological effects can include post-traumatic stress symptoms, confusion, fear about infection, frustration, boredom, inadequate supplies, inadequate information, financial loss and stigma, all with the possibility of long-lasting effects [5,6,7]. However, previous research, with a focus on consequences and social distancing or quarantine due to different reasons, revealed that some of the key factors that can mitigate the negative consequences of a quarantine are: social interaction and support, keeping people informed and reducing boredom and improving communication [5,8,9]. Social interaction and support are found to serve as a ‘buffer’ between stress and mental health [10,11], and keeping people well-informed can help moderate the fear that most people feel when exposed to a worrying infectious disease [12]. Boredom and isolation can cause a lot of distress during a quarantine. However, in addition to having practical advice on coping and stress-management techniques, having a mobile phone and using social media to be connected with family and friends has shown to be essential in reducing the effects of immediate anxiety and longer-term distress [13,14,15]. Thus, social media could play a crucial part in communication in a crisis, allowing people to stay in touch and reassure their loved ones, as well as to stay informed on local and global events.

However, using social media is not without its drawbacks. Since the COVID-19 outbreak, social media has played a pivotal role in disseminating information about the pandemic as well as a tool to overcome the constraints of social distancing while providing solidarity and support resources for those in lockdown situations. This was first evident in China after the outbreak, where the emphasis was on using social media in such a way that provided an opportunity to communicate the reasons for the quarantine, and provided reassurance and practical advice in order to prevent rumors and panic [16]. At the same time, the World Health Organization (WHO, Geneva, Switzerland), the Centers for Disease Control and Prevention (CDC), scientific journals and health organizations worldwide used different social media platforms to inform and guide the public in the pandemic. However, in addition to the official guides, social media platforms such as Facebook, Twitter, YouTube and Instagram provided access to an extraordinary amount of information that was contradictory to what was official, and perhaps amplified rumors and misinformation. This is because social media algorithms account for people’s preferences and facilitate content promotion, and therefore, the spread of information [17]. This can then shift the worldview, can construct different social perceptions and can change the narratives, particularly when issues are controversial [18]. The paradigm shift as a result of this can influence policymaking, political communication and overall, the development of a public debate [19]. Despite the efforts of social media platforms to direct users to the WHO, CDC and websites of local health authorities for the correct information [20], evidence suggests that heightened media exposure of crisis information is associated with increased distress, worry and long-term impaired physical and mental health [21].

Furthermore, recent literature suggests that the frequency, length and diversity of media use and exposure were positively associated with depressive symptoms and both unspecific and COVID-19-related anxiety [22]. In addition, Scopelliti et al. [23] point out that mild use of social media can have a positive effect on people while excessive social media use can increase the negative consequences of social media use.

While there have been a number of studies exploring various aspects of social media during COVID-19, such as social media for rapid knowledge dissemination [24], the impact of COVID-19 on psychosocial consequences [25] or the growth of misinformation via social media during the pandemic [18], this is the first study to our knowledge looking at the qualitative experiences and challenges of using social media amid the pandemic by the three named nations. Thus, this study provides an in-depth analysis of the experiences and challenges associated with using social media by the multinational population in a time window after the start of the pandemic where most people relied on social media as a means for gathering information and as a tool for staying connected to family, friends and work.

The aim of this study was to explore the trans-national participants’ experiences and challenges in using social media while in lockdown or shelter-in-place during the initial stages of the COVID-19 pandemic.

## 2. Methods

The consolidated criteria for reporting qualitative research (COREQ) was used to guide the structure of this paper [26].

### 2.1. Study Design

This qualitative study was part of a wider trans-national online survey about mental health, wellbeing, loneliness and the use of social media during the pandemic [27]. The cross-sectional survey was made available between April and May 2020. Invitations to take part in the survey were placed on social media such as Twitter, Facebook and Instagram in Norway, the USA, UK and Australia. Data were collected for approximately a 3–4 week period in each country. Each country had a landing site for the survey at the researcher’s respective universities; OsloMet—Oslo Metropolitan University, Norway; University of Michigan, USA; University of Salford, UK; and the University of Queensland, Australia.

### 2.2. Participants and Setting

The initiator of the project was AØG from OsloMet, Norway; however, each participating country/university had a project lead adhering to local ethical approvals. Participants in the study were the general population in the participating countries. The participants fulfilled both of the inclusion criteria: (1) be of age of 18 or over and (2) speak Norwegian or English. Any respondent who did not meet the above criteria was excluded. The introductory information accompanying the survey explained the purpose and anonymity of the study, and consent was obtained by completing the first page of the survey. The study was thereby quality-assured and approved by OsloMet (20/03676) and the regional committees for medical and health research ethics (REK, 132066) in Norway, reviewed by the University of Michigan Institutional Review Board for Health Sciences and Behavioral Sciences (IRB HSBS) and designated as exempt (HUM00180296) in the USA, by the University of Salford Health Research Ethics (HSR1920-080) in the UK, and the University of Queensland (HSR1920-080; 2020000956) in Australia.

### 2.3. Questionnaire

The online questionnaire was developed and designed by the lead country in collaboration with the partner countries. Different survey platforms were used in the countries: Webform (Nettskjema) in Norway, Qualtrics in the USA and the Bristol Online Survey platform (onlinesurveys.ac.uk) in the UK and Australia. Webform, Qualtrics and the Bristol Online Survey are secure survey-development tools [28,29,30]. The survey had 41 questions, including demographic variables, COVID-19-relevant items and well-known reliable and validated questionnaires related to mental health, wellbeing and quality of life, in addition to an open-ended question. Only the responses from the qualitative open-ended question are presented in this paper (see research question below). All open responses were reviewed in detail to identify common themes. The Norwegian version of the questionnaire did not include the open-ended question; thus, the Norwegian population was therefore excluded from the results and discussion.

### 2.4. Survey Question

1. *During this COVID-19 pandemic, what challenges have you experienced in using social media?*

### 2.5. Analysis

All data were pooled together. Two researchers (M.S., J.L.) independently coded the data to minimize subjectivity. All data were analyzed using a thematic analysis approach [31]. Thematic analysis is a rigorous method consisting of 6 phases, providing structure for the data to be organized, coded, and themes to be identified. First, the two researchers read all the data twice in order to get familiarized with it. For phases two and three, the researchers generated initial codes and then searched for themes among the codes independently. The researchers then met to discuss their findings and to extract the main themes. After this, phase 4 and 5 followed, where M.S. and J.L. met and reviewed (checked if the themes worked in relation to the code extracts and the overall data set) and defined and named (further analysis to refine the specifics of each theme) the themes before agreeing on the final names of the key themes and sub-themes (see Table 1). A report (phase 6) on the findings was then presented to all researchers and discussed.

#### Rigor

Trustworthiness and rigor are often the criteria by which qualitative studies are judged, meaning that integrity and competence have to be demonstrated within a study [32]. These criteria were maintained by the analysis being conducted by two researchers independently. M.S. and J.L. met at three points (after the third, fifth and sixth step) to discuss findings and settle any potential disputes—there were none in this case. Rigor with sampling was ensured through maximum variation sampling (diverse sample) to make the data ‘information-rich’ [33].

## 3. Results

### 3.1. Demographics

Out of a total of 3810 participants that responded to the wider survey, 1991 participants answered the qualitative question and were included in qualitative analysis (Table 2).

### 3.2. Themes Emerged

Three major themes emerged at the final point of analysis: Emotional/mental health, Information and Being connected. The themes were similar across the participating countries. Further to the main themes, separate sub-themes unfolded (see Table 1). The themes and sub-themes were inter-connected rather than independent (see Figure 1).

### 3.3. Emotional and Mental Health

The data suggested that the participants were concerned about the impact of social media on their emotional and mental health. Four subthemes of emotional/mental health were identified in the coding process. The categories that were consistently identified by the participants as being associated with their mental health were: anxiety and depression, anger and frustration, panic and fear and/or isolation and loneliness. The foundations for these subthemes are described separately alongside the verbatim quotes. Some of the quotes illustrate more than one sub-theme simultaneously. See Figure 1 for interconnectedness between the subthemes in this theme. For example, if we look at information overload as a subtheme for information, this subtheme was very closely linked with anger, frustration and fear and panic that resulted in having an impact on emotional/mental health.

#### 3.3.1. Anxiety and/or Depression

Some participants described that, due to social media being one of the main sources of information and due to the vast information and notifications from it during the lockdown or shelter-in-place, they had heightened feelings of anxiety or depression or it was a trigger for already-existing mental health issues. Others often had to negotiate with themselves if using social media to stay connected could warrant the heightened anxiety that came from using social media in the first place. ‘*… I have long relied on social media (esp. Facebook) for staying connected with people I do not get to see in person. I have also enjoyed social media content (esp. Instagram and Twitter) as a mode of entertainment, as it connects to my hobbies and pursuits. Now, all three of those platforms heighten my anxiety…*’ (P2591) *‘It has caused my depression to get worse, so I’ve absolutely had to be very strict about how I use it in order to avoid nasty panic attacks*’. (P1333).

#### 3.3.2. Anger and/or Frustration

A subgroup of participants described feeling angry or frustrated when using some of the social media platforms. Some of this anger and frustration was directed towards those that are spreading misinformation. Others were directing their anger and frustration towards their governments and how they dealt with the pandemic. Some were worried about the increased anger expressed in many of the social media posts that later translated into expressing anger towards the people in their households or social networks. Other participants were frustrated that not everyone complied with the COVID-19 lockdown and shelter-in-place rules. ‘*Frustration at seeing others share inaccurate or dangerous information*’. (P463), *‘and others also get so angry at those who constantly break the covid rules and from friends and myself I know this anger was let out at those closest*’. (P1418).

#### 3.3.3. Panic and/or Fear

Participants observed that social media has been filled with information that is creating panic or fear and is often spread by the different media outlets. Some described that they had to stop using social media for a period of time due to the panic it was creating for them. Others expressed that social media in the pandemic has brought on fear relating to health, finances and the feeling that this is the end of the world. Overall, many found it challenging to avoid the increased panic in social media due to the pandemic. ‘*In the early stages I came off (social media) for a few days because everyone was panicking and it was making me wobble*’. (P1045), ‘*Panic. Fear for my health. No income. Feeling this is the end of the world*’. (P2993).

#### 3.3.4. Loneliness and/or Isolation

A group of participants highlighted the sense of isolation or loneliness that was exacerbated by using social media or relieved by social media. One participant said that they had to use social media because they are lonely but using social media did not help them feel less lonely, as computers are not people. On the other hand, there were participants saying they need to use social media, as, without it, they would feel lonely and disconnected. ‘*I use it more because I’m lonely, but it does not actually help. Computers are not people*’. (P3739), ‘*However, I need social media because otherwise I feel lonely and disconnected*’. (P3623). 

### 3.4. Information

The second theme was about information. Four separate subthemes were identified: information overload, the volume of misleading information, the polarity of information and the pressure of having to constantly ‘be available’ and not being able to switch off.

#### 3.4.1. Information Overload

A portion of the participants described that they felt excessiveness of variety of information. For example, some felt that there was information overload related to the surviving of the pandemic, which was overwhelming and anxiety-inducing. A few had to learn to turn off social media so they can cope with the information overload. Others felt exhausted from spending too much time online in order to stay connected with their social networks. Despite this, many felt the compulsion to continue using social media, as underscored by some of the respondents: ‘*Information overload, compulsive viewing of FB, lowers mood but continue to spend too much time on it*’. (P1442).

#### 3.4.2. Volume of Misleading Information

The majority of participants commented about the volume of misleading information on social media. Some of the participants highlighted the difficulty of accessing the correct information online due to it being mixed with rumors and non-factual information. Others commented on the sheer volume of information that was misleading and sensationalized by mainstream media and the large focus on negativity. *‘**There is often too much fake information that can be misleading. There is also too much focus on the negatives of COVID.’* (P1548)

A few participants commented on the frustration about the self-proclaimed ‘experts’ that were responsible for spreading misinformation and rumors without a scientific basis. ‘*Too many „experts” without a clue spreading fake news and rumours. People too quick to become keyboard warriors!*’ (P971).

#### 3.4.3. Polarity

Participants noticed that the information on social media was very polarized. Not only was there an information overload, but there was an overload of polarity around the available information. Some participants noted that the majority of the social media information was very negative and ranged from trolling to negative political news or negative emotions of self or others (friends and family). Others, however, noted that there was overwhelming ‘think positive’ information and challenges that were not realistic and frustration-inducing. Many participants were seeking out more positive information or were starting to adopt strategies to filter the negative information. ‘*Negative emotions regarding loved ones’ encouragement of the politicism of a global pandemic. Negative emotions regarding the extremism of people’s political views. Negative emotions regarding the perceived selfishness of people who would rather enjoy their life at the expense of others than stay home and wait it out in order to preserve life.**’* (P2870) *‘Pick your information streams to stay informed if correct and accurate information. Seek motivational, gratitude sources to maintain positive outlook, to remind that humans are adaptive and can change to new circumstances*’. (P 21).

#### 3.4.4. Cannot Switch Off

Participants described that there was too much external pressure to keep using social media for work and social activities, and it was hard to ‘switch off’, which added extra stress and made them feel overwhelmed. Other participants stressed that the constant pressure to ‘be available’ to use video calls and online activities created more stress for them. This differed from the subtheme of staying connected or even addicted, as the inability to switch off was about the external pressure/expectations to stay connected rather than the internal drive or compulsion to stay on social media or to stay on social media for the purpose of staying connected. ‘*You need to use them more than ever in order to stay in touch, learn, undertake social activity* e.g., *exercise. As a result you feel you’re constantly using technology and never switch off from it*’. (P733), ‘*… feeling pressure to be „available” because we don’t have a „legitimate reason” to be busy*’. (P2898). 

### 3.5. Being Connected

This theme was around striving for communication or being connected via social media and efforts for learning.

#### 3.5.1. Connection/Communication

There were a variety of experiences associated with connection and communication. Some participants found social media as a tool to help them reach out and connect with their larger social networks and made them feel as a part of the community or without social media they would have felt lonely and disconnected. Others, however, found it difficult to feel connected to their friends and family due to being unable to see them in person for long periods or to fully see their facial expressions and body language. ‘*It has been helpful in connecting with others, especially those in my larger social network who I do not reach out to regularly, and with feeling like part of a community*’. (P2603), ‘*But I need social media because otherwise I feel lonely and disconnected*’. (P3623).

#### 3.5.2. Learning

This subtheme had a range of feelings associated with learning and being connected. A portion of the participants said that they had to learn to use the different social media platforms or how to navigate them in a short period of time, which caused them to feel stressed and confused. Others found learning to communicate via social media challenging, particularly when having to make decisions while unable to properly read body language or hear the other side. A few found the experience of learning the new platforms helped them offset patience and had turned it into an opportunity for new competencies. *‘**I had to learn quickly about new online platforms due to work. I only used Skype and Whatsapp… Now I use Zoom, Microsoft Teams, Attend Anywhere and Pow Wow (phone conference with multiple people). Learning about this online platforms in a short period of time has been stressful and confusing but getting there.’* (P356).

## 4. Discussion

The primary aim of this study was to explore the experiences and challenges of using social media during the current COVID-19 pandemic. The strengths of this study are the diverse sample of participants and systematic and in-depth method of analysis of the experiences and challenges associated with using social media by the multinational population in a time where most relied on social media for gathering information and staying connected to family, friends and work. To the authors’ knowledge, this is the first transnational study to qualitatively explore the experiences and challenges of using social media during the COVID-19 pandemic. The findings have given a clear insight into what the key challenges around social media use during the pandemic are.

Our findings about the contributing factors of social media use during the COVID-19 pandemic on emotional and mental health are in line with previous findings related to the epidemics of Ebola and SARS, where social media amplified the feelings of anxiety and fear [34,35] as well as recent findings from the current pandemic where high social media exposure was linked with higher prevalence of mental health problems [36,37]. For instance, we found that a range of emotions, such as anxiety, depression, frustration, anger, panic, fear and loneliness, was amplified from using social media during the quarantine. This finding could be seen as a contradiction with the findings of a recent rapid review, where the suggested strategies of having internet access and activating social media were recommended to reduce panic, isolation and stress among those in a quarantine [4]. However, we do not know if the amplified emotions recognized by the participants in this study are a direct result of using social media or are perhaps a result of being in a quarantine, which we know amplifies negative emotions and triggers further mental health issues [4,38]. Alternatively, it could be that during the pandemic, participants spent a. longer time passively using social media. This is in line with other research suggesting that in a crisis, there is a tendency of an increasing time of social media use [30,38]. Evidence suggests that the prolonged passive use of social media (scrolling through a social media news feed) is associated with depression [39].

From our findings, we can also see that many of the amplified emotions identified by the participants were related to finding information or navigating through the volume of misleading and polarized information, particularly when social media during the pandemic was bombarded with misinformation and negative and confusing information [36]. This was also in line with the literature on a ‘fight against infodemic’ that poses a serious problem for public health globally [23,40] and as a result of an enormous amount of information coming from different sources and often with no valid foundation, today’s news spreads very quickly and through multiple channels, almost like viruses [41].

Nonetheless, we know from previous evidence that when the public is exposed to negative and confusing information, they start to express negative feelings such as fear, panic and anxiety themselves [42]. We also know that obtaining correct information in a timely manner should reassure people when dealing with a crisis or when in a quarantine and should therefore lessen the feelings of fear, panic and distress [5,13,14]. However, a recent study suggests that regardless of having correct information in a timely matter on social media, the distribution of accurate and non-accurate information in social media has similar spreading patterns [18]. These spreading patterns are mainly driven by the specific interaction patterns of users engaging with a certain topic on the specific media platforms. Thus, it could be that work needs to be done on understanding the social dynamics behind the urge for content consumption. This, in return, could help with designing better communication models for social media that could account for social behavior and therefore provide more efficient communication strategies in a crisis situation and reduce the amplified negative emotions. Lastly, the participants expressed some positive urges to be connected and to learn new ways of communication via social media. Many participants reported that they used social media to be connected with their family and friends. While at times, this way of connecting was not perfect or as good as in-person, it helped many of the participants remain connected or feel less isolated than if social media was unavailable. This is in line with previous findings where people in crisis felt more supported and optimistic about the future when social media was extensively involved [43]. This brings an important consideration about the reasons why social media is used by the participants and about its ways of passive and active use. Previous studies suggest that when people use social media in an active way to connect with other people, share and reveal emotions, by uploading either text, videos or pictures, social media becomes a tool of emotional regulation, and therefore, their experiences of using social media are positive and increase the feelings of being connected or supported [44,45]. In contrast, when people use social media in a passive way, they often have amplified negative feelings [46]. Thus, social media use has both positive and negative effects on us. The negative effects may be perceived as more dominant in the early stages of this crisis (rapid spread, confusion about which measures should be taken and the consequences they will have for the individuals). This could mean that as the pandemic progresses, people will start to use social media in positive ways and deliberately seek information that can bring more positive emotions, which in return can help with coping with the crisis situation.

## 5. Limitations

There are several limitations associated with this study. There might have been a selection bias in that those who held particularly strong feelings about the topic could have been more likely to participate in the study. Additionally, most of the participants were from the UK, so there might not be equal representation of all themes, and the results may have been different if the Norwegian population had been included. This has to be taken into consideration in later research. The survey title used in the advert for participants was: ‘The effects of COVID-19 on mental health and quality of life’, so awareness or need to improve mental health and quality of life may be greater in the sample population when compared to the general population or those that participated could be more comfortable discussing mental health issues. The Norwegian sample was not asked the qualitative question, which warrants future research. A substantial proportion of participants did not answer the qualitative question. The missing data may include a mix of people who had not experienced any challenges with social media use during the COVID-19 lockdown or shelter-in-place and those who had but did not respond. Future research that first asks a filter question may help to reduce the level of missing data due to no challenges to report. The wording of the qualitative question was ‘what challenges have you experienced in using social media?’, which may have led to more negatively focused responses than if the wording was different, for example, asking more broadly about ‘experiences’ instead of ‘challenges’. Nonetheless, while the question asked participants to report any challenges, still, some positives were reported. The positives and how social media has been helpful to the community throughout the pandemic may be equally important to understand in addition to challenges faced, and therefore warrants investigation in future research.

## 6. Implications for Policy and Practice

This study suggests that using social media while in shelter-in-place or lockdown could have a negative impact on the emotional and mental health of the population across the participating countries. Some of these negative effects on mental health can be longer-term and policymakers and public health organizations should consider measures to respond to the likely surge of ill mental health after the lockdown and shelter-in-place. While anxiety and depression were some of the most common themes from the study, there were a number of participants seeking positive or helpful information via social media. This could be an opportunity for the development of future therapeutic interventions that could be delivered or partially delivered via social media and, in return, mitigate the amplified negative feelings of the population. In addition, appropriate attention is needed for those that either had previous ill mental health or those for whom the shelter-in-place and lockdown triggered more serious mental health issues.

A lot of the anger, frustration, panic and fear were related to the content and volume of information or misleading information about the pandemic on social media. Public health and the governments should look at ways to use social media to provide relevant, coherent and correct information in a timely manner with increased transparency and clear pathways of accessing that information, particularly in a peak phase of crisis as previously found in Macao [47]. However, they should also consider the social dynamics in their countries and how that knowledge can be used to improve communication. One suggestion is utilizing the dialogic communication theory [48] and creating a dialogic loop [49], where the government and the general public engage in a negotiated interchange of ideas and opinions that promote the realization of mutual satisfaction and creation of common meaning [50]. In addition to combating misleading information on social media, its psychological impacts needs to be addressed.

Our finding that emotions and mental health pose as one of the key challenges imply that support in this area is urgently required. The addressing of mental health needs has been raised as an integral part of responding to COVID-19 [51,52]. Examples of specific strategies include strengthening mental health care in the community and investing in health workers [2]. While the immediate response to COVID-19 has been focusing on ensuring the capacity of clinicians and health care workers in the infectious disease field, we need to, at the same time, ensure that the communities are also able to gain support from health care workers in the mental health field, such as psychologists, mental health nurses and social workers.

## 7. Conclusions

The findings revealed that using social media while in shelter-in-place or lockdown could have a negative impact on the emotional and mental health on some of the population in the participating countries. It also revealed that the amplified negative emotions were often related to navigating through the volume of misleading and confusing information. Communication with the public should be strengthened by unambiguous and clear messages and clear communication pathways. Nonetheless, for some participants, using social media in the pandemic helped them remain connected or feel less isolated than if social media was unavailable. For the others, we should be looking at alternative ways of staying connected during a similar crisis. Although these findings might not represent views from the whole general population, to support policy and practice in strengthening mental health care in the community, social media could be used to deliver practical advice on coping and stress management.

## Figures and Tables

**Figure 1 healthcare-09-00735-f001:**
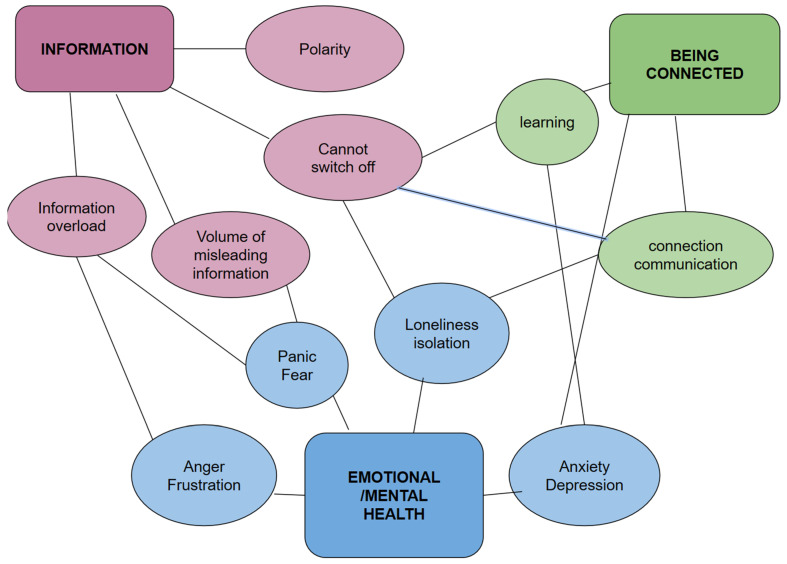
Matrix of interconnectedness of themes and subthemes.

**Table 1 healthcare-09-00735-t001:** Themes and subthemes.

Themes	Subthemes
Emotional/mental health	Anxiety, depression
	Anger, frustration
	Panic, fear
	Loneliness, isolation
Information	Information overload
	Volume of misleading information
	Polarity
	Cannot switch off
Being connected	Connection, communication
	Learning

**Table 2 healthcare-09-00735-t002:** Participants characteristics.

	Total	UK	USA	Australia
	*n* = 1991	*n* = 1013	*n* = 801	*n* = 177
Age group				
18–29	18.5%	16.4%	18.4%	30.5%
30–39	18.5%	18.1%	19.2%	17.5%
40–49	20.7%	24.3%	17.2%	15.8%
50–59	20.8%	22.8%	18.7%	18.6%
60+	21.6%	18.5%	26.5%	17.5%
Gender				
Female	80.9%	85.4%	75.9%	77.4%
Male	17.0%	13.2%	21.3%	19.2%
Other/prefer not to say	2.2%	1.4%	2.9%	3.4%
Area of residence				
Rural or Farming	7.2%	5.7%	10.1%	2.3%
Small town	22.2%	16.8%	31.1%	13.0%
Medium-Sized City	37.0%	37.9%	40.8%	14.7%
Large City	33.6%	39.6%	18.0%	70.1%
Education				
High school or below	8.7%	7.7%	9.5%	10.8%
Technical/Associate degree	19.2%	23.2%	13.1%	24.4%
Bachelor’s degree	32.4%	34.1%	32.7%	21.6%
Master’s/Doctoral degree	39.6%	35.0%	44.7%	43.2%
Live with someone				
Yes	82.0%	83.0%	80.2%	84.7%
No	18.0%	17.0%	19.8%	15.3%
Currently in work				
Yes, in a Full-time job	45.9%	46.2%	46.6%	40.7%
Yes, in a Part-time job	22.1%	23.8%	17.4%	34.5%
No	32.0%	30.0%	36.1%	24.9%
How often have you used social media after COVID-19 pandemic started?				
Weekly or less	1.6%	1.7%	1.3%	2.3%
A few times a week	3.0%	3.8%	1.8%	3.4%
Daily	24.5%	26.2%	20.6%	32.8%
Several times daily	70.9%	68.3%	76.4%	61.6%

## Data Availability

The datasets used and/or analyzed during the current study are available from the corresponding author on reasonable request.

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
