# Peer review of "Mental Health, Information and Being Connected: Qualitative Experiences of Social Media Use during the COVID-19 Pandemic from a Trans-National Sample"

_healthcare, 2021, doi:10.3390/healthcare9060735_

Round 1

Reviewer 1 Report

Dear Editor,
The authors have added the proposed changes. It is a simple but useful article, and from a methodological point of view it is quite correct. For my view it can be accepted already.
Best regards

Author Response

 Dear reviewer. Thank you for your time. Your review and comments have improved this manuscript significantly.

Reviewer 2 Report

I am happy that our comments have helped the authors in improving their paper. I am satisfied with the revised version of the manuscript. Please be careful to the references list, some references are present in the manuscript but not in the references list (e.g. Scopelliti et al., 2021 https://doi.org/10.3390/ijerph18041879

Please check also for guidelines about in-citations and references. The authors have reported in alphabetical order the references list and have given a number to each reference, but these numbers are not present in the manuscript.

Author Response

Dear reviewer. Thank you for your time. Your review and comments have improved this manuscript significantly. We have added the following reference to the reference list: Scopelliti, M., Pacilli, M. G., & Aquino, A. (2021). Tv news and covid-19: Media influence on healthy behavior in public spaces. International journal of environmental research and public health18(4), 1879.

We believe that the numbering of the references is automatic with the uploading system as we didn’t number the references. Thus, we have now numbered the references in the text and resolved the problem.

Reviewer 3 Report

This version has addressed most of the issues. Apart from the following minor editing issues, I am happy to recommend acceptance. Congratulations to the authors for their hard work!

1. The numbering in the Results section seems confusing. I think 3.9-3.11 should be under sub-section 3.8, right? Shouldn't it be 3.8.1, 3.8.2, etc.?

2. Please use consistent capitalisation of COVID - sometimes Covid is used.

Author Response

Dear reviewer. Thank you for your time. Your review and comments have improved this manuscript significantly.

Point 1. The numbering of the sections is automatic but we have corrected this now and put  note for editors.

Point 2. Thank you. We have corrected the capitalisation of COVID throughout the article.

This manuscript is a resubmission of an earlier submission. The following is a list of the peer review reports and author responses from that submission.

Round 1

Reviewer 1 Report

Thank you for providing me with the opportunity to review this manuscript. Upon review, I unfortunately believe that there are fundamental issues with the manuscript's rationale, hypotheses, methodology and results that need to be addressed before publication.

This study present seems like a trend report, the “Discussion and conclusion” is a data analysis report; it seems not any academic contribution. I think this is not appropriate.  The Introduction should not present a hypothesis. This chapter describes your background, motivation and research purpose. Although the author refers to many studies, much of the literature you reviewed might be considered to be irrelevant. 

Reviewer 2 Report

In the present manuscript, the authors explored in a cross-national study the experiences and challenges of social media use during the Covid-19 pandemic. The topic is interesting and the paper is well written. 

My main concern regards the lack of originality compared with another recent work, entitled "Mental health, quality of life, wellbeing, loneliness and use of social media in a time of social distancing during the COVID-19 outbreak. A cross-country comparative study" (https://pubmed.ncbi.nlm.nih.gov/33689546/). The conclusions of both manuscripts are identical. The main difference I can see between the two manuscripts is the quantitative vs qualitative method. I ask the authors to clarify the difference among these studies, by underlining the novelty of their study compared with that published in Mental Health Journal. 
In the Introduction, I recommend the authors citing more recent papers about the media and covid. Specifically, Bendau et al. (2020, Associations between COVID-19 related media consumption and symptoms of anxiety, depression and COVID-19 related fear in the general population in Germany. Eur. Arch. Psychiatry Clin. Neurosci, 1–9, doi:10.1007/s00406-020-01171-6) showed that the frequency, duration, and diversity of media exposure were positively associated with symptoms of depression and both unspecific and COVID-19-related anxiety. The authors also identified a critical and intriguing threshold between mild and moderate use of media. In my opinion, this last information could be really interesting, giving that can be the key point to differentiate between positive and negative themes that emerged in the thematic analyses of the manuscript I am revisioning. I mean that mild use of social media can help people to be in contact with others until a critical point when the excessive use of social media can increase the negative consequences of social media use.  The differentiation between the consequences of mild and moderate use of media has been confirmed by another recent study about media and covid (Scopelliti, M., Pacilli, M. G., & Aquino, A. (2021). TV News and COVID-19: Media Influence on Healthy Behavior in Public Spaces. International journal of environmental research and public health, 18(4), 1879. https://doi.org/10.3390/ijerph18041879) Please add this suggestion to the manuscript with the suggested references.
The theme called “overload information” is surely interesting. I think that need to be inserted in the theoretical framework of infodemy, namely the dissemination of an enormous amount of information coming from different sources and whose foundation is often not verifiable. Just like viruses, news today spreads very quickly and through multiple channels (Eysenbach, G. 2020. How to fight an infodemic, https://doi.org/10.1016/S0140-6736(20)30461-X)
Please give more details about the methodology you used in the selection of themes and sub-themes of open-ended questions.
Please give more details about Figure 1, especially the relationships among the variables.

Reviewer 3 Report

This paper presents the results of a survey conducted in the UK, USA and Australia about how and why people use social media during the COVID-19 pandemic. A total of 1991 responses were analysed using qualitative thematic analysis. The findings summarise three themes: mental health, information and being connected. Policies and practical recommendations were given at the end of the paper to promote the use of social media to address the needs of the public.

This paper is very timely and addresses a trendy research gap in COVID-19. The research design and analysis have been done adequately and the paper is well written. I have a few comments/questions as follows:

  1. Page 3: is the name of the institution which approved the ethics application missing? 
  2. Page 3: there were 42 questions in the survey but it seems that only 1 question was reported. Does this paper only extract this question and report on this? If yes, the authors need to explicitly mention in the paper that only some parts of the survey are used.
  3. The paper does not actually analyse the surveys from Norway so please remove Norway from the abstract and the rest of the paper.
  4. The distribution of the three countries is skewed. How to ensure that the themes emerging from the analysis are universal, and not biased towards/dominated by a particular country?
  5. Page 8: The distinctions between the themes "cannot switch off" and "being connected" are not obvious. Also, is "cannot switch off" the same as "addicted to social media"?
  6. Page 11 Implications for Policy and Practice: when discussing about the government's use of social media, I recommend citing the below articles and elaborate your findings based on these papers.

Thanks for conducting this meaningful research and also the review opportunity of this paper. I look forward to your revision.

[1] Pang, P. C.-I., Cai, Q., Jiang, W., & Chan, K. S. (2021). Engagement of Government Social Media on Facebook during the COVID-19 Pandemic in Macao. International Journal of Environmental Research and Public Health18(7), 3508. https://doi.org/10.3390/ijerph18073508

[2] Chen, Q., Min, C., Zhang, W., Wang, G., Ma, X., & Evans, R. (2020). Unpacking the black box: How to promote citizen engagement through government social media during the COVID-19 crisis. Computers in Human Behavior110, 106380. https://doi.org/10.1016/j.chb.2020.106380